# Structural MRI-Based Schizophrenia Classification Using Autoencoders and 3D Convolutional Neural Networks in Combination with Various Pre-Processing Techniques

**DOI:** 10.3390/brainsci12050615

**Published:** 2022-05-09

**Authors:** Roman Vyškovský, Daniel Schwarz, Vendula Churová, Tomáš Kašpárek

**Affiliations:** 1Faculty of Medicine, Institute of Biostatistics and Analyses, Masaryk University, Kamenice 3, 625 00 Brno, Czech Republic; schwarz@iba.muni.cz (D.S.); churova@iba.muni.cz (V.C.); 2Department of Psychiatry, University Hospital Brno, Masaryk University, Jihlavska 20, 625 00 Brno, Czech Republic; tkasparek@fnbrno.cz

**Keywords:** schizophrenia, classification, 3D CNN, autoencoders, voxel-based morphometry, deformation-based morphometry, deep learning

## Abstract

Schizophrenia is a severe neuropsychiatric disease whose diagnosis, unfortunately, lacks an objective diagnostic tool supporting a thorough psychiatric examination of the patient. We took advantage of today’s computational abilities, structural magnetic resonance imaging, and modern machine learning methods, such as stacked autoencoders (SAE) and 3D convolutional neural networks (3D CNN), to teach them to classify 52 patients with schizophrenia and 52 healthy controls. The main aim of this study was to explore whether complex feature extraction methods can help improve the accuracy of deep learning-based classifiers compared to minimally preprocessed data. Our experiments employed three commonly used preprocessing steps to extract three different feature types. They included voxel-based morphometry, deformation-based morphometry, and simple spatial normalization of brain tissue. In addition to classifier models, features and their combination, other model parameters such as network depth, number of neurons, number of convolutional filters, and input data size were also investigated. Autoencoders were trained on feature pools of 1000 and 5000 voxels selected by Mann-Whitney tests, and 3D CNNs were trained on whole images. The most successful model architecture (autoencoders) achieved the highest average accuracy of 69.62% (sensitivity 68.85%, specificity 70.38%). The results of all experiments were statistically compared (the Mann-Whitney test). In conclusion, SAE outperformed 3D CNN, while preprocessing using VBM helped SAE improve the results.

## 1. Introduction

Schizophrenia represents a severe neuropsychiatric disorder that affects nearly 20 million people worldwide [1]. As treatment can help more efficiently if an appropriate antipsychotic is prescribed in the early stage, a fast diagnosis is crucial [2]. However, there is no known objective marker, such as a finding in a blood test or a shape in the brain anatomy, to conclusively diagnose schizophrenia, so current diagnosis relies on an interview with a psychiatrist. During the interview, the patient shows the symptoms they suffer from, such as hallucinations or delusions. Still, there is no objective diagnostic result to confirm the diagnosis and be available for presentation to the patient and their family.

In past decades, there have been efforts to take advantage of computational methods as tools for neuroimaging data analysis to determine the neurobiological nature of psychiatric diseases. In the early stages, statistical methods were used to detect differences between the brain of a healthy person and a patient diagnosed with, for example, schizophrenia [3,4,5,6,7], Alzheimer’s disease [8], or borderline personality disorder [9,10,11] at the group level. Manual region-of-interest (ROI) methods were used to measure the size of a specific structure in the brain and compare it between patients and healthy subjects. However, this method is unsuitable for comparability between laboratories due to vaguely defined borders between anatomical structures [8]. Recently, fully automated brain morphometry methods have emerged that focus not on a single structure but the whole brain or its tissues. The method, termed voxel-based morphometry (VBM), has become a standard pipeline that processes image data from an anatomical MRI scan into a statistically comparable image [12,13]. The pipeline starts with registration and resampling into a standardized space, continues with segmentation into brain tissues–white matter (WM), gray matter (GM), and cerebrospinal fluid (CSF), and ends with Gaussian kernel smoothing, resulting in gray matter density (GMD) features. A statistical map with *p*-values can be counted on features and describes the statistical significance of the differences between groups in each voxel. VBM-based approaches revealed structural changes in schizophrenia, particularly in the left superior temporal gyrus region [3], decrease in GM volume in the thalamus, middle orbital gyrus, inferior frontal gyrus, and other structures [6], and decline in GM bilateral occipital lobe, left orbital frontal cortex and other structures [7], as well as GM reduction in the ventral cingulate gyrus and several regions of the medial temporal lobe in patients with Borderline Personality Disorder [9]. These findings suggest that schizophrenia is manifested by morphological abnormalities in many brain structures. Another brain morphometry method that uses anatomical MRI scans is deformation-based morphometry (DBM). DBM tracks the shifts, turns, and other movements of the brain tissues while registering the image on a standard brain template. The resulting features are represented by local volume changes (LVC), which can also be statistically evaluated at the voxel level.

Findings obtained using various automatic brain morphometry methods suggest that there are differences between healthy and diseased groups that are revealable by statistical analysis of various features extracted from brain image data. Despite these successful results based on a group-level analysis, there is a lack of a supportive diagnostic tool that would reveal with specific probability whether a patient is affected by a brain disorder or not. In comparison with statistical methods, classifiers work at the subject level. Many classification methods have been used to solve this task. Statistics-based methods such as linear discriminant analysis and its modifications have been applied to imaging data in schizophrenia research [14], whereas machine learning methods, which, in comparison with statistical methods, learn hidden patterns in the images, often iteratively based on the training dataset, proved to be a good way to classify neuroimaging data. There have been a lot of studies using machine learning to classify neuropsychiatric diseases. Some authors used support vector machines (SVM) to classify schizophrenia with features extracted from MRI data using wavelet transform [15] or from fMRI data using independent component analysis [16]. Other features were extracted by calculating the Pearson correlation of WM integrity in diffusion-weighted images [17] or features extracted and selected with SVM alone [18]. Other authors took advantage of ensemble learning, which employs multiple classifiers trained on different data or features, to provide a decision by voting [19,20,21]. Although the classification of schizophrenia is still challenging, some machine learning methods have been very successful in diagnosing some neurodegenerative diseases, such as Alzheimer’s disease. Savio et al. used backpropagation, radial basis function, and probabilistic neural network [22], while Huang et al. [23] used VBM and artificial neural networks (NNs).

The current success of deep learning builds on the historical development of NNs, which began with the perceptron [24] and the multi-layer perceptron (MLP) [25]. The MLP is a commonly used classifier. However, it may be overfitted with a growing number of neurons and layers. A very deep architecture can easily learn the information, such as subject class, and classify well on a training group, but the similarly successful results may not be achieved on test data. A better way to learn NNs is to go layer by layer while controlling the correctness of the learning. Such models are called stacked autoencoders.

An autoencoder [26] is a single-layer neural network in which the input information is the same as the expected target; thus, the hidden layer of neurons tries to reconstruct the input [27]. The single layers are then cascaded to create a deep stacked autoencoder network, which is finally tuned by a backpropagation algorithm. Autoencoders are used to classify many brain diseases. Employing the leave-site-out method for validation, Zeng et al. [28] diagnosed schizophrenia with a deep discriminant autoencoder network on functional connectivity MRI in a multi-centric study, reaching an accuracy of 85% and 81%. Moussavi-Khalkhali and Jamshidi [29] used sparse AE to extract features for classifying Alzheimer’s disease vs late and mild cognitive impairments and controls using multinomial logistic regression. In addition to the standard AE (82.88% accuracy), they used partially cascaded AE that works with information from each layer as an input to the classifier (92.44% accuracy). They also used denoising autoencoders. Nayak et al. [30] employed a modified autoencoder (SRVFL-AE) to classify MRI with multiclass brain abnormalities such as degenerative disease, stroke, tumor, infectious disease, and normal brain, with high accuracy exceeding 95%. Mendoza-Léon et al. [31] developed a supervised switching autoencoder based on the principle of reconstructing two output channels. The better-reconstructed channel defined the input class. They had single slices taken from structural MRIs of patients with Alzheimer’s disease and achieved an accuracy higher than 87.5%. Deep wavelet autoencoders helped to detect cancer in the brain [32]. Pinaya et al. [33] used deep autoencoders to classify schizophrenia with AUC-ROC 0.611-0.751.

Another effective method for training deep architectures is the convolutional neural network (CNN) [34,35]. The convolutional neural network has its 3D variant for imaging data with its 3D nature, such as MRI. The 3D CNN consists of several types of layers. The main type is a convolutional layer that counts the convolution operation between the kernel and the input image or output of the previous layer. The convolution operation brings two advantages. The CNN works with a much smaller number of weights than traditional MLP because the weights for each convolution kernel are shared for each kernel position during convolution. It strongly reduces the number of network parameters, which can help to tackle overfitting. Compared to MLP, SVM, Naïve Bayes, or other classifiers, the convolution works with information about the position of neighboring voxels. Each convolution’s result serves as the activation function’s input, which is necessary to turn the network into a nonlinear model, and the standard for CNN is RELu (Rectified Linear Unit). Two other types of layers follow: the batch normalization (BN) layer and the max-pooling (MP) layer, several layers with fully connected neurons follow, and typically, the output layer for the classification task is SOFTMAX.

CNN has found application in many medical image classification and schizophrenia applications, delivering good results. Wang et al. [36] worked with functional MRI and took advantage of the dilated 3D convolution, achieving an accuracy of 80%. They suggest preprocessing the data by slice timing, realigning, normalization, and smoothing. Three-dimensional CNN was used in [37] on structural MRI preprocessed by VMB, reaching 79.27% accuracy and 70.98% accuracy on the validation set, respectively. Hu et al. [38] trained 3D CNN on structural and diffusion MRI, preprocessed by skull stripping, registration, segmentation, and modulation, with an accuracy of 79.27% and pre-trained 2D CNN with an accuracy of 72.41%. Oh et al. [39] classified schizophrenia patients on a multi-centric structural MRI dataset (n = 873) with an accuracy of 97%, but the accuracy achieved on the validation dataset was lower than 70%. Campese et al. [40] compared 3D CNN, 2D CNN and SVM on structural MRI data (preprocessed by VBM), with 3D CNN outperforming the other two. Therefore, the authors suggest that 3D convolution can capture spatial information correctly. The authors also applied the augmentation technique in CNN and noted a positive effect on accuracy.

The two previously mentioned deep architectures (CNN and AE) were combined by Oh et al. [41]. They used a 3D convolutional autoencoder to diagnose schizophrenia spectrum disorders based on task-based fMRI (the task was to evaluate positive, negative, and neutral images with an unpleasant or neutral reaction). In doing so, they achieved an accuracy of around 84%. The autoencoders served as a tool for initializing the weights before the final tuning of the convolutional network.

Since the quality of the classifiers is directly affected by the dataset’s quality, one of the main challenges is using an appropriate image preprocessing pipeline. There are various feature extraction and selection methods that can successfully extract only important and well-discriminative information from the image. Voxel and deformation-based morphometries can serve as feature extraction methods, as can wavelet transform [15] or independent component analysis [16] in the case of fMRI data. In this paper, the advantage of morphometry methods is exploited. Voxel-based morphometry [12] prepares segmented and smoothed tissues, whereas deformation-based morphometry preprocesses the whole brain and prepares information, as some shifts are necessary for template fitting. Since CNNs are known to classify the original image without feature selection, the third dataset used in this paper uses T1-weighted images preprocessed using only the necessary steps to remove the skull and non-brain parts. As previously described, a whole-brain approach without much preprocessing could be advantageous because schizophrenia manifests inhomogeneously in the brain.

Deep learning methods are state-of-art classifiers known for their ability to extract appropriate information for classification. However, morphometries as feature extraction methods used to play a crucial role in brain preprocessing, where shallow classification methods were applied and enabled to prepare information for classifiers, improving the classification pipeline. Even nowadays, authors tend to use preprocessing [36,37,38,40]. To the best of our knowledge, various morphometric methods have not yet been investigated and compared in combination with deep learning classifiers, such as autoencoders and 3D CNN-based models, and the ability of these methods to properly extract features alone has not been compared with morphometry preprocessing methods.

In this paper, several research questions are asked and answered experimentally. First, SAE and 3D CNN deep learning methods were employed to classify schizophrenia patients vs controls from our dataset. Second, CNN, in particular, does not need much preprocessing. Therefore, two morphometric methods, VBM and DBM, were applied and assessed to see if this feature extraction helps classify deep learning-based models compared to minimally preprocessed MRI. The third goal was to explore how architectural changes, such as the number of layers and neurons, affect each feature extraction outcome. Fourth, features extracted by the three methods were combined in all possible ways, and the impact on classification results was observed.

## 2. Materials and Methods

### 2.1. Schizophrenia Patients and Healthy Controls

All the subjects were patients at the University Hospital Brno. The dataset was 52 patients in the first episode of schizophrenia (SZ) and 52 healthy control subjects (HC). The entire dataset consisted of only male patients with median (min-max) age: SZ 22.9 (17–40), HC 23.0 (18.2–37.8). The dataset groups were age-matched to limit the effect of gray matter volume reduction with aging [42]. All patients were diagnosed according to ICD-10 (International Classification of Diseases). Their blood and urine samples were collected and tested for toxicology, hematology, and biochemistry to exclude patients with abnormal findings. None of the subjects had a family or personal history of Axis I psychiatric disorder. All subjects signed informed consent, and the study was approved by the ethics committee [43].

A previous study [44] involved a subset of 39 patients and an equal number of controls. Another of our previous studies [14] comprised 49 patients and 49 healthy controls recruited from the same clinical workplace. The numbers of subjects differed due to prospectivity in the study design. The data collected from this study were also used in our previous paper focused on training a multi-layer perceptron with random subspace ensembles [21].

All subjects were scanned using a 1.5 T Siemens Symphony MRI machine with the following parameters: the sagittal tomographic plane thickness was 1.17 mm, the in-plane resolution was 0.48 mm × 0.48 mm, the 3-D field of view contained 160 × 512 × 512 voxels, IR/GR sequence, TR was 1700 ms, TE was 3.93 ms, TI was 1100 ms, the flip angle was 15°, and FOV was 246 × 246 mm [43].

### 2.2. Feature Extraction

There are many brain imaging modalities, each carrying a different type of information, which together provide comprehensive data for data modeling based on multidimensional statistical methods and machine learning techniques. In this experiment, structural MRI data provided information for deep neural network models. The data were preprocessed using three different standardized pipelines supplying different image features: (i) image registration with skull stripping and two automated morphometry methods, (ii) voxel-based morphometry, and (iii) deformation-based morphometry.

The first dataset, later referred to as the T1 dataset, consisted of features corresponding to T1-weighted intensities in structural MRI images. The images were spatially normalized, the skull was stripped, and the CSF was removed using SPM8 software (Statistical Parametric Mapping package for MATLAB: http://www.fil.ion.ucl.ac.uk/spm/, accessed 11 March 2022) to preserve only those brain parts with white and gray matter. The second dataset was preprocessed using the VBM pipeline [45], later referred to as the GMD dataset, implemented in the SPM8 toolbox (http://dbm.neuro.uni-jena.de/vbm/, accessed 11 March 2022). The pipeline, described in detail in [21], included intensity inhomogeneity correction, spatial normalization, GM segment extraction and its modulation with the determinants of the Jacobian matrices computed during the nonlinear registration, and final smoothing using an 8 mm FWHM Gaussian kernel. The third dataset was preprocessed using DBM, which involved identical correction and spatial normalization steps as for VBM. Instead of working with only the GM tissue segment, the DBM pipeline transforms the brain images into local volume changes resulting from the determinants of the Jacobian matrices extracted from vector displacement fields obtained by the high-dimensional registration algorithm [46]. This preprocessed dataset of local volume change features is later referred to as the LVC dataset.

### 2.3. Experiment 1: Autoencoders

The first experiment focused on the use of stacked autoencoders. The pipeline of the classification process, consisting of simple feature selection, SAE with specific configurations, and validation methods, is shown in Figure 1. The first part of the algorithm is the selection of voxels that could fundamentally improve the success rate of the networks and tackle the problem of the time-consuming learning process. The selection metric was the Mann-Whitney test, i.e., a non-parametric method, so no assumption about the data distribution had to be met. After putting aside the testing samples, the Mann-Whitney test was applied to each voxel of the brain images voxels at a specific location in each patient’s brain, creating one group, and voxels at the same location in healthy brains created a second group. The application of the tests on the whole brain resulted in a probability map of *p*-values. These values were ranked from lowest to highest so that only the most discriminating voxels could be selected. The results are not interpreted as statistically significant differences between the two groups–the healthy and patients–but serve as a tool to select only those voxels that could lead to good classification results. The selection was performed using a threshold value related to a predefined *p*-value, i.e., an uncorrected significance level. By thresholding, voxels were divided into two groups. The first group served as the set of features for learning classifiers and is referred to as the feature pool (FP). The second group was omitted in further analysis. The parameter examined was the size of the feature pool, which was set to 1000 and 5000 voxels. Since the feature pool of 5000 voxels did not bring significant classification improvement and was very computationally expensive, the feature pool was not further enlarged.

Once the feature pool was defined, stacked autoencoders could be learned on selected voxels to classify healthy controls and patients. The SAE-based classifiers have many parameters that need to be set, so the size of the feature pool and two other network settings were explored. These were the number of layers and the number of neurons. The latter always decreased from the input to the output layer, so the information going through the network was continually compressed to eventually provide a small feature space for the classification by the SOFTMAX layer. The number of neurons was set equal to or smaller than the number of inputs. Although it would have been very informative to compute all combinations of layers and neurons within layers, the experiment would have been very demanding on computational resources, which were limited. Therefore, trends were observed rather than looking for the best results across all possible combinations of network architectures.

Stacked autoencoders with the following settings were used for classification based on all the three datasets: each layer was learned using 100 epochs, the supervised SOFTMAX layer was learned in 1000 epochs, the regularization parameters were left as default (L2 regularization was equal to 0.004, sparsity regularization was equal to 4, and sparsity proportion to 0.15), and the training was fine-tuned. The training process was validated using 10-fold cross-validation, where 10% of the dataset was put aside in each fold before the features were selected and the network trained. After the last fold was performed, the complete testing data set consisted of 104 subjects, and each subject was classified by a network that was not trained on that subject to achieve unbiased results. The entire training process was repeated ten times to tackle the variability caused by random weight initialization. Finally, average accuracy metrics were assessed.

### 2.4. Experiment 2: 3D Convolutional Neural Network

Like autoencoders, 3D convolutional neural networks were used to solve the problem of classifying all three types of features, i.e., intensity with minimal preprocessing, gray matter density, and local volume changes (see the classification process in Figure 1). Since the convolutional neural network can extract important features by itself, no other feature selection method, such as the Mann-Whitney test, was applied in this experiment. Furthermore, by omitting this selection step, no information about the position of neighboring voxels was lost, so the images were treated as 3D structures, which is crucial for convolution operations in CNNs.

The parameters examined were the number of convolutional layers and convolutional kernels. The deep nature allows the neural network to learn and compose the structure of complex features across layers; the more layers, the more space to work with the features. However, the size of the network also affects the risk of overfitting, and a very deep CNN may not provide benefits, especially if the dataset is not large enough. The convolution kernel size was set to 3 × 3 × 3 voxels by default, and the number of convolutional kernels increased from the input to the output layer. The convolutional layer was always followed by the batch normalization, RELu, and max-pooling layers. The particular 3D CNN architectures examined can be seen in Table 1. The other parameter settings were as follows: the input image dimension was 121 × 145 × 121 × 1, where the last dimension was the number of channels (grayscale channel only); the stride was set to 3, the dilation factor was 1, the max-pooling dimension was the same as filter dimension (3 × 3 × 3 voxels), the number of neurons in the fully connected layer was 10 hidden and 2 output neurons, the dropout = 0.5, the initial learning rate was 0.0001 for the Adam algorithm, the data were shuffled in each epoch, and the mini-batch size was 10. Training was performed on batches (subsets) of data to estimate the gradient over the training set. The more data there is in a batch, the better the gradient estimation, and it is more efficient than calculating the gradient for each subject separately [47]. The training was controlled by the validation set, which consisted of 10% of the training data, and was stopped after 1000 epochs or when all of the following criteria were met: the minimal number of epochs was 100, accuracy on the validation set was decreasing, training accuracy was greater than 90%, and validation accuracy exceeded 75%. These criteria enabled the 3D CNN to learn with high accuracy in a reasonable amount of time.

The training process was validated using 10-fold cross-validation in the same way as for SAEs, and since the weights were initialized randomly and the learning process was random due to shuffling in both network types, the entire learning process of each architecture was performed ten times. Therefore, all assessment metrics (accuracy, sensitivity, and specificity) were averaged, and their mean values are presented here.

### 2.5. Experiment 3: Combined Features

The final experiment focused on combining the input data to investigate whether this step would improve accuracy. T1, GMD, and LVC features were thus combined in all possible ways, i.e., T1 + GMD, T1 + LVC, GMD + LVC, and T1 + GMD + LVC were put together as input data. In the case of autoencoders, feature selection based on the Mann-Whitney test was applied to each dataset separately, and particular feature pools were concatenated to serve as input data for autoencoders with various architectures. Regarding 3D CNNs, features were combined using a Directed Acyclic Graph (DAG), allowing multiple inputs in the next layer. The T1, GMD, and LVC features were processed separately by 3D convolutional, batch normalization, RELu and max-pooling layers in their own channels. Then the outputs were combined and processed using fully connected layers. This way, the DAG architecture enabled assessing whether any feature combinations could improve accuracy. Since learning the 3D CNN classifier based on the combination of the two or three features was computationally very intensive, we decided to explore only CNNs of five and seven convolutional layers, which gave good results when single modality was used. Adding more layers was beyond our computational possibilities.

## 3. Results

This section summarizes the results obtained in all experiments. First, the results of the first and third experiments (comprising autoencoders) are described, followed by the results of all classification experiments with convolutional neural networks. Autoencoder experiments were performed in Matlab R2020a on a computer with 2× Dual-Core AMD Opteron (tm) Processor 2220 2.80 GHz and 32 GB RAM. The 3D CNNs experiments were performed in Matlab R2020a on a computer with 2× Intel^®^ Xeon^®^ CPU E5-2640 2.5 GHz and 64 GB RAM.

### 3.1. Autoencoders

The results counted for the combination of two feature pool sizes and six SAE architectures are summarized in Table 2 for all three features.

#### 3.1.1. T1

The classification results achieved on the minimally preprocessed dataset, denoted as T1, did not exceed 60% of the accuracy metric. The best result was an accuracy of 56.35% (sensitivity 65.77%, specificity 46.92%) using a feature pool with 5000 features and 500 neurons. The model with a feature pool of size 1000 voxels had the best accuracy of 54.13% when an autoencoder with 50 neurons was trained. The accuracy of the other architectures varied between 48.65–54.52%.

#### 3.1.2. GMD

The autoencoders learned on the GMD performed up to 13% better than those learned on T1. The best accuracy was achieved on the architecture with FP = 1000 and 500-50 hidden neurons, and its results were 69.62% (sensitivity 68.85%, specificity 70.38%). Other architectures that outperformed 67% were 500, 1000-500-100, 1000-500-10 based on FP = 1000. Regarding FP = 5000, architecture with 1000-500-100 reached the highest accuracy (66.25%).

#### 3.1.3. LVC

The third input feature type, LVC, achieved better results than T1 but not as good as GMD. The average accuracy varied between 52.12% and 63.37%. More features added to the feature pool helped classification since accuracy on FP = 1000 was between 52.12% and 56.63% compared to 55.1% up to 63.37% on FP = 5000.

### 3.2. Autoencoder Combination

After the SAEs were trained on a single feature, an experiment followed in which all the features were combined. The results are summarized in Table 3.

#### 3.2.1. T1/GMD

This combination of features improved classification compared to the T1-based classifier but worsened the results compared to the classifiers learned on GMD. The models with FP = 1000 achieved accuracy between 52.5% and 55.29%, and the models learned on FP = 5000 had accuracy between 54.62% and 58.46%.

#### 3.2.2. T1/LVC

The combination of T1 and LVC features yielded the worst results in all the combinations of the two feature types. The accuracy of architecture with FP = 5000 and 50 hidden units was 56.15%. As in the previous combination, a bigger feature pool improved the results (accuracy > 53.08%) compared to FP = 1000 with all average accuracies < 52.79%.

#### 3.2.3. GMD/LVC

The last combination of the two features had the best results of all. The architecture with FP = 5000 and 500 units yielded an accuracy of 61.54% (sensitivity 69.62%, specificity 53.46%). Accuracy based on FP = 5000 was again higher than on FP = 1000 with all average accuracies > 59.14%. However, the results were worse compared to autoencoders learned on GMD only.

#### 3.2.4. T1/GMD/LVC

The combination of all three features did not improve the outcomes as expected. The accuracy ranged from 49.71% to 59.62%. When comparing the best models from both feature pools, the results on the bigger feature pool were more than 4% better.

### 3.3. 3D Convolutional Neural Networks

Seven architectures were created to observe the ability of the 3D CNN to perform our classification task. The number of layers was set to range from three to nine filters, i.e., with an increasing number of filters. The experimental results (average accuracy/average sensitivity and average specificity) are summarized in Table 4.

#### 3.3.1. T1

The 3D CNNs trained on the first dataset achieved accuracy higher than 60% when networks with more than five layers were used. The growing number of filters positively affected the results, which can be seen in architectures with the same number of layers. Overall, the higher number of filters added more than 5% to the accuracy of architectures of the same depth. The most successful architecture had seven layers and an accuracy of 60.39%.

#### 3.3.2. GMD

Training on preprocessed data using VBM resulted in better outcomes than the T1 dataset. The highest average accuracy was 63.08% when a seven-layer network was used. The shallowest network, the 10-50-100, failed in classification, and the other architectures had accuracies between 56.73–63.08%. The larger number of filters improved the results similarly to the previous T1 data case.

#### 3.3.3. LVC

3D CNNs combined with LVC input data failed to classify schizophrenia patients and healthy subjects correctly. All defined architectures achieved results between 45.58–53.75%. No trends of improving the results were observed. The 3D CNN with three layers and LVC input data achieved the poorest results in all three datasets.

### 3.4. DAG 3D CNN

Due to the computational cost, only two architecture experiments were performed. The first had five layers, and the second had seven layers. The results for the combination of input datasets are shown in Table 5.

#### 3.4.1. T1/GMD

The combination of T1 and GMD in 3D CNN reached an accuracy of 60.67% for five filter layers and 62.31% for seven filter layers. In both cases, the accuracy of 3D CNN improved compared to the scenario using T1 data but was slightly worse compared to GMD data use.

#### 3.4.2. T1/LVC

The combination of T1 and LVC reached an accuracy of 54.42% for 3D CNN with five filter layers and 51.64% for seven filter layers. In both cases, the result was worse than for CNN models based on T1 alone, but the shallower 3D CNN outperformed it when LVC alone was used.

#### 3.4.3. GMD/LVC

The combination of T1 and LVC reached an accuracy of 58.65% for 3D CNN with five filter layers and 59.14% for 3D CNN with seven filter layers. This combination of input data did not bring any improvement either. The results were better if LVC alone was used, but pure GMD-based classifiers reached even higher accuracy.

#### 3.4.4. T1/GMD/LVC

The combination of all three feature types reached an accuracy of 58.46% for 3D CNN with five filter layers and 59.90% for 3D CNN with seven filter layers. GMD-based models or a combination of T1 and GMD yielded better outcomes.

### 3.5. Statistical Comparison

Finally, the architectures with the best result for each dataset (T1, GMD, LVC) and models (SAE, 3D CNN) were statistically compared using the Mann-Whitney test. Accuracies that resulted from repeating all experiments 10 times were used as the data that entered the statistics, and the specific compared architectures were FP = 5000 and 500 hidden neurons (T1), FP = 1000 and 500-50 hidden neurons (GMD), FP = 5000 and 500 hidden neurons (LVC) for stacked autoencoders, and 50-100-150-200-250-300-350 architecture for 3D CNN and each feature type. The comparison omitted feature combination since its performance did not exceed classifiers based on a single feature type. The results can be seen in Table 6.

## 4. Discussion

This paper contributes to the research and explores the possibility of computer-aided classification of schizophrenia using deep learning algorithms. Its main contribution is the investigation of the influence of input data on classification results and thorough testing and comparison of two modern deep learning classifiers. Stacked autoencoders and 3D CNNs were applied to classify schizophrenia patients and healthy controls. The classification was based on MRI data preprocessed using three different pipelines to extract the three following types of features: T1 with simple registration on a template and skull stripping, GMD with the use of voxel-based morphometry, and LVC using deformation-based morphometry. The research focused on three experiments: (1) two deep learning classifiers were trained to classify schizophrenia based on various data preprocessing pipelines, (2) two important network architecture parameters were explored: depth and number of neurons in each layer, and (3) feature extraction methods were finally combined in all possible ways to train the deep learning classifiers.

The main limitations of the study are small size of the sample dataset (which is difficult to collect, especially in neuroscience), data-related unavailability of validation data from another center, insufficiently extensive but also a very time-demanding investigation into the combination of possible parameters and model architectures–SAE and 3D CNNs have many parameters to be set, such as number of layers, size of filters, number of filters, size and frequency of pooling layer, number of layers with fully-connected neurons and their quantity, which makes the exploration of all possible settings almost impossible. Instead of the whole grid of parameter combinations, only trends of some main parameters were investigated. Finally, a method that could show the impact of brain regions on classification is missing, which was not the aim of this study.

The stacked autoencoders achieved different accuracy for each type of feature. The best result obtained using T1 features was an accuracy of 56.35%. In contrast, for LVC features, an accuracy of up to 63.37% was achieved (in both cases with only a shallow network consisting of two layers with 500 hidden and two output neurons and FP = 5000). Deeper architectures did not lead to any accuracy improvements for T1 input data cases. Such low accuracy rates suggest that the simplest image preprocessing (T1) is unsuitable for stacked autoencoders. In the case of LVC features, a bigger feature pool (FP = 5000) allowed the network to classify better than a smaller feature pool (FP = 1000). The accuracies obtained using LVC features ranged from 56.63% (FP = 1000 with 100-50-10 architecture) to 63.37% (FP = 5000 and 500 architecture). A similar trend in accuracy was observed for T1 features. There, FP = 1000 and 50 architectures led to an accuracy of 54.13%, while FP = 5000 and 500 architectures led to an accuracy of 56.35%. However, both feature extraction methods (simple registration and skull stripping and DBM) did not seem to provide sufficient information for the given classification task, so the stacked autoencoders performed poorly. Better results were achieved with features extracted using the VBM pipeline. GMD enabled SAE-based classifiers to achieve accuracies of up to 70%. The best architecture (FP = 1000 and 500-50) achieved an average accuracy of 69.62% (sensitivity 68.85%, specificity 70.38%). This time, a smaller feature pool (FP = 1000) led to better results, with only the deepest architecture (1000-500-100-50-10) failing, with the networks classifying almost all subjects as patients, which resulted in an average sensitivity higher than 90% and an average specificity lower than 13%. Depending on the specific architecture, a larger pool of features (FP = 5000) led to accuracies between 50.77% and 66.25%. However, the 100-50-10 and 1000-500-100-50-10 architectures also failed to classify well, as in the case of FP = 1000. In conclusion, when only one dataset source was used for SAE classification, GMD provided the most informative features because its results were better than those using T1 and LVC. A smaller feature pool was better when using GMD. However, when T1 and LVC were used, more features helped increase the accuracy of the results. This suggests that GMD might have better discriminative information hidden in fewer features while not carrying much noise. The main finding of these SAE experiments is that both VBM and DBM pipelines brought a noticeable advantage for SAE classification compared to SAE accuracy results on the minimally preprocessed dataset, suggesting that SAE cannot extract image characteristics by itself, and preprocessing steps are needed.

Other authors have used autoencoders to classify patients with schizophrenia. Zeng et al. [28] achieved an accuracy of >80% with a deep autoencoder discriminant network. They reached better accuracy results, but worked with fMRI data on a larger sample size (n = 734) that might have affected the results. Comparable classification results to ours (AUC-ROC 0.611-0.751) were achieved by Pinaya et al. [33] with a deep autoencoder and a normative method.

The combination of feature types entering SAEs did not bring any improvement. The most successful combination was GMD and LVC on a larger pool (FP = 5000) and 500 architecture with an average accuracy of 61.54% (sensitivity = 69.62%, specificity = 53.46%). However, the results were worse than SAEs performed on GMD and LVC only. Any trends related to the size of the feature pool or SAE architecture were hardly observable. The combination of all three feature types brought an improvement only when compared to the results obtained by SAEs trained on T1 data, which we considered inappropriate.

According to the results obtained, 3D CNNs trained on T1 features needed at least five convolutional layers to train. Architectures with three convolutional layers only achieved around 50% or less accuracy, whereas 3D CNNs with more than five convolutional layers reached up to 60%. Unfortunately, more layers did not help to improve the results. In any case, 3D CNNs seem to have outperformed SAEs trained on the same data (T1). Architectures with five or more convolutional layers led to an accuracy almost 4% higher than the best SAE architecture (FP = 5000 and 500 architecture). The reason may lie in the 3D structure of filters and the whole-brain approach. Regarding the 3D CNNs trained on GMD, the results were slightly better than in the case of the T1 dataset. In terms of accuracy, the best architecture (50-100-150-200-250-300-350-10) was identical to the T1 data but with almost up to 3% better results. The improvement may suggest that schizophrenia is more manifested in the gray matter, which is extracted from the brain in contrast with T1 or DBM, which is a whole-brain method. However, the trends were similar to the previous dataset. The three-layer networks did not perform well. Compared to SAE, different datasets did not help improve the results, suggesting that 3D CNN can learn to extract features from images themselves and without preprocessing. On the other hand, the results were not as good as in the case of SAEs, suggesting that SAE can work more effectively with a smaller sample size, or that the preprocessing method of feature selection helped reduce noisy voxels from the images, or a combination of both. The 3D CNNs trained on the DBM (LVC) preprocessed data somehow failed in classification, the accuracy being less than 54%. We assume that the DBM pipeline deformed the relations between brain structures, which are useful for CNN classification. In summary, based on our exploratory work and experimental results, it is obvious that the 3D CNNs are not as powerful a tool for classifying schizophrenia based on MRI data compared to SAE, at least for such a small dataset. A small dataset may not provide enough training data for a model as complex as the 3D CNN. However, it is highly challenging to collect large enough datasets in neuroscience. VBM and DBM preprocessing did not affect the results as much as in the case of SAEs.

DAG 3D CNN networks did not outperform any model based on a single feature dataset. Input information combined from two or three data sources with the same sample size of patients and controls probably contributed to overfitting the models in the training phase only because it exacerbated the curse of dimensionality–too many features for too many subjects. Thus, feature combination is rather useless with such as small sample size.

In the context of schizophrenia classification, 3D CNNs have been employed by other researchers. Campese et al. [40] used pre-trained 3D CNNs (LeNet, VNet, and UNet combined with SVM) to classify VBM preprocessed data. They achieved an accuracy of 86.3% (n = 101) and 71.63% (n = 176) on two independent datasets, respectively. The dilated 3D CNN achieved >80% accuracy in classifying 2300 fMRI EPI images [36], but the dataset consisted of only 46 subjects. Moreover, the modality differed from that used in this paper. Oh et al. [41] used 3D CNN combined with task-based fMRI data (n = 82) reaching ~84% accuracy. The same dataset as in this paper was used in our previous work. The first paper [21] explored random subspace ensembles of multi-layer perceptrons (MLP) and SVMs, which were learned on the VBM and DBM data and their combination. The achieved accuracies were 73.12% (MLP) and 73.51% (SVM). The second paper [20] presented the results on SVM trained on selected voxels using two-sample t-tests with an accuracy not exceeding 64%. These results show that the results achieved in this paper are worse compared to shallow methods trained on the same dataset and suggest that shallow methods can achieve better results with smaller sample sizes.

There are studies with well-performing classifiers whose sample dataset does not exceed n = 130 (recommended by [18]), which may have affected the results. These are the following: (i) [17], which used neural network-based classifiers (ANN, LVQ, RBF) with a 100% accuracy (n = 20), (ii) [16], which incorporated NN on fMRI data and reached an accuracy of 75.6% (n = 69), and (iii) [15], which used SVM on structural MRI achieving an accuracy over 71% (n = 104). All of the above models outperformed our results. Our drawback was the small sample size (N = 104).

A final statistical comparison of the results shows that the SAE-based classifier trained on GMD features outperformed all other investigated classifiers and input feature datasets. On the other hand, the SAE-based classifier on T1 data and 3D CNN-based on LVC data performed statistically significantly worse than most of the other classifiers. The results suggest that it is worthwhile to use GMD resulting from the widely accepted VBM pipeline combined with SAE-based classification.

## Figures and Tables

**Figure 1 brainsci-12-00615-f001:**
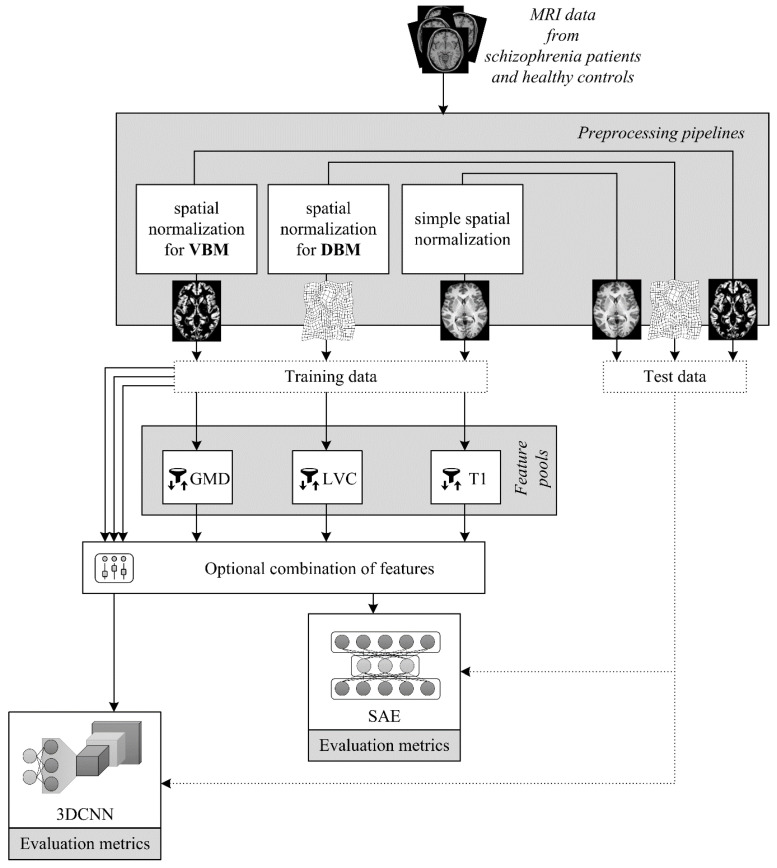
Scheme of the performed experiments with SAE-based and 3D CNN-based classifiers trained on different types of brain imaging features. There are three distinct features extracted using different image preprocessing pipelines–two complex pipelines taken from automated morphometry methods (VBM, DBM) and a simple one that includes only registration and skull-stripping operations. The features are pooled using univariate testing, sorting, and thresholding in the case of the SAE classifier, whereas all features are taken in with the 3D CNN classifier. The experiments further involve changes in the architectures and the optional combination of different feature types.

**Table 1 brainsci-12-00615-t001:** The 3D CNNs architectures used. The numbers in the chart represent the number of kernels in a particular layer. The dashes stand for the following: batch normalization, RELu, and max-pooling layers. There is a fully connected layer at the end of each network with 10 hidden and 2 output neurons.

3D CNN Name	3D CNN Architecture
CNN-1	10-50-100
CNN-2	20-40-60-80-100
CNN-3	20-50-100-150-200
CNN-4	5-10-20-40-60-80-100
CNN-5	10-20-30-40-50-60-70-80-90
CNN-6	50-100-150-200-250-300-350
CNN-7	50-100-150-200-250-300-350-400-450

**Table 2 brainsci-12-00615-t002:** Results for SAE learned on the three types of features (T1, GMD, and LVC). The metrics listed represent the accuracy (sensitivity/specificity) in percentage.

Feature Pool	Neurons	T1 [%]	GMD [%]	LVC [%]
1000	500	48.75 (58.27/39.23)	67.6 (64.42/70.77)	56.44 (58.27/54.62)
1000	50	54.13 (47.88/60.38)	65.48 (61.15/69.81)	54.71 (55.96/53.46)
1000	500-50	48.65 (46.73/50.58)	69.62 (68.85/70.38)	52.12 (58.27/45.96)
1000	1000-500-100	52.31 (46.54/58.08)	68.37 (67.69/69.04)	52.98 (54.81/51.15)
1000	100-50-10	52.12 (39.81/64.42)	67.12 (67.12/67.12)	56.63 (52.5/60.77)
1000	1000-500-100-50-10	50.87 (39.42/62.31)	51.44 (90.19/12.69)	56.15 (60/52.31)
5000	500	56.35 (65.77/46.92)	50.77 (96.92/4.62)	63.37 (62.88/63.85)
5000	50	54.52 (58.85/50.19)	50.77 (97.31/4.23)	57.4 (51.92/62.88)
5000	500-50	54.13 (71.73/36.54)	59.04 (54.04/64.04)	61.06 (55.77/66.35)
5000	1000-500-100	52.79 (54.81/50.77)	53.08 (54.23/51.92)	58.65 (55.96/61.35)
5000	100-50-10	50.58 (30/71.15)	65 (64.04/65.96)	57.31 (45.19/69.42)
5000	1000-500-100-50-10	51.25 (39.42/63.08)	66.25 (67.12/65.38)	55.1 (49.42/60.77)

**Table 3 brainsci-12-00615-t003:** Results for SAE learned on all combinations of the three types of features (T1, GMD, and LVC). The metrics listed represent the accuracy (sensitivity/specificity) in percentage.

Feature Pool	Neurons	T1/GMD [%]	T1/LVC [%]	GMD/LVC [%]	T1/GMD/LVC [%]
1000	500	53.17 (55.19/51.15)	50.77 (49.23/52.31)	58.46 (58.46/58.46)	49.71 (47.31/52.12)
1000	50	52.5 (47.12/57.88)	48.56 (53.08/44.04)	53.46 (57.5/49.42)	50.38 (49.04/51.73)
1000	500-50	55.29 (50.19/60.38)	49.42 (44.62/54.23)	59.23 (64.62/53.85)	51.44 (52.69/50.19)
1000	1000-500-100	53.46 (47.5/59.42)	52.79 (50.77/54.81)	60 (61.73/58.27)	52.21 (47.12/57.31)
1000	100-50-10	52.98 (36.15/69.81)	51.15 (47.69/54.62)	56.63 (66.15/47.12)	53.94 (45/62.88)
1000	1000-500-100-50-10	52.5 (37.31/67.69)	52.02 (40/64.04)	59.9 (62.5/57.31)	55.19 (50.96/59.42)
5000	500	57.12 (59.23/55)	56.06 (59.62/52.5)	61.54 (69.62/53.46)	59.62 (55/64.23)
5000	50	57.21 (55.38/59.04)	56.15 (57.12/55.19)	60.87 (67.69/54.04)	55.19 (54.42/55.96)
5000	500-50	56.35 (47.31/65.38)	54.42 (53.65/55.19)	60.58 (66.92/54.23)	55.87 (58.08/53.65)
5000	1000-500-100	54.9 (54.04/55.77)	55.1 (57.69/52.5)	59.9 (59.81/60)	55.1 (52.69/57.5)
5000	100-50-10	58.46 (43.65/73.27)	56.06 (50/62.12)	59.13 (60.19/58.08)	58.27 (56.92/59.62)
5000	1000-500-100-50-10	54.62 (47.88/61.35)	53.08 (53.85/52.31)	59.9 (66.15/53.65)	56.92 (63.46/50.38)

**Table 4 brainsci-12-00615-t004:** Results for 3D CNNs learned on the three types of features (T1, GMD, and LVC). The metrics listed represent the accuracy (sensitivity/specificity) in percentage.

3D CNN Architecture	T1 [%]	GMD [%]	LVC [%]
10-50-100	43.37 (43.85/42.86)	42.40 (46.15/38.65)	45.58 (41.15/50.00)
20-40-60-80-100	54.62 (57.5/51.73)	61.15 (62.31/60.00)	52.98 (56.15/49.81)
20-50-100-150-200	60.15 (63.25/57.05)	61.65 (61.11/62.18)	51.92 (52.12/51.73)
5-10-20-40-60-80-100	50.29 (48.85/51.73)	56.73 (57.89/55.58)	51.73 (52.69/50.77)
50-100-150-200-250-300-350	60.39 (61.54/59.23)	63.08 (63.85/55.58)	53.75 (51.15/56.35)
10-20-30-40-50-60-70-80-90	53.27 (52.89/53.66)	59.52 (59.81/59.23)	52.89 (51.54/54.23)
50-100-150-200-250-300-350-400-450	60.19 (60.77/59.62)	62.6 (60.00/65.19)	50.86 (52.78/48.93)

**Table 5 brainsci-12-00615-t005:** Results for 3D CNNs learned on all combinations of the three types of features (T1, GMD, and LVC). The metrics listed represent the accuracy (sensitivity/specificity) in percentage.

3D CNN Architecture	T1/GMD [%]	T1/LVC [%]	GMD/LVC [%]	T1/GMD/LVC [%]
20-40-60-80-100	60.67 (59.81/61.54)	54.42 (52.69/56.15)	58.65 (57.69/59.62)	58.46 (60.00/56.92)
50-100-150-200-250-300-350	62.31 (60.19/64.42)	51.64 (49.23/54.04)	59.14 (58.27/60.00)	59.90 (57.69/62.12)

**Table 6 brainsci-12-00615-t006:** *p*-values of the Mann-Whitney test comparing the accuracies obtained from 10 repetitions of the experiments for both classifiers (SAE, CNN) and all input feature types (T1, GMD, LVC). Significant results (α = 0.05) in favor of the classifier, listed on the left, are marked with an asterisk (*). Those in favor of the classifiers listed above are marked with two asterisks (**).

Classifier-Input Data	GMD-SAE	LVC-SAE	T1-3D CNN	GMD-3D CNN	LVC-3D CNN
T1-SAE	3.6580 × 10^−4^ **	0.0018 **	0.0811	0.0018 **	0.0799
GMD-SAE	-	0.0098 *	0.0031 *	0.0031 *	1.7265 × 10^−4^ *
LVC-SAE	-	-	0.4459	0.9694	2.3313 × 10^−4^ *
T1-3D CNN	-	-	-	0.4919	0.0254 *
GMD-3D CNN	-	-	-	-	2.3313 × 10^−4^ *

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
