# Peer review of "Structural MRI-Based Schizophrenia Classification Using Autoencoders and 3D Convolutional Neural Networks in Combination with Various Pre-Processing Techniques"

_brainsci, 2022, doi:10.3390/brainsci12050615_

Round 1

Reviewer 1 Report

The manuscript by Vyškovský applies two different deep learning approaches (stacked autoencoders and 3D- convolutional neural networks [3D-CNN]) to build classifiers for discriminating patients with a first episode of schizophrenia from healthy controls by means of three types of brain images derived from T1 MRI scans. The subject of the manuscript is of current interest for both the target population and the methods used. However, I have some methodological issues and doubts listed below.
1.- Deep learning methods, due to the very large amount of parameters to be estimated, are typically applied to samples ranging from the hundreds to the millions (i.e. big data) where these methods excel. Here an N = 52 is used for each group. I acknowledge that recruiting first episode patients is difficult, but it I wonder if with such small samples the algorithms are just underperforming, and a “shallow” learning algorithm such as the SVM, random forest, GPC, lasso … would have equally o better performed. Including the results of one of them would enrich de manuscript.
2.- The author claim that both samples are matched for age, but what about gender? Since males and females differ in their T1 images, if the proportion of males is different in both samples the authors may have unknowingly fit classifiers for gender instead of disorder.
3.- Before fitting the autoencoders, the authors carried out a selection of most informative voxels through their significance in M-W tests. From the text, it may seem that they used the whole sample of patients and controls to make this previous selection of voxels. If this is the case, that would lead to clear positive biases in the estimates of validation accuracy (as data from validation individuals would have been used for the selection of voxels). This needs to be clarified.
4.- The authors carried out fittings for models with different number of layers and units (filters in CNN) as a kind of exploratory search for the best model, but how do they decide on the rest of hyperparameters? (number of epochs, size of batches, regularization, function types, …). Did they use the same data for selecting hyperparameters and evaluating the success of models?
5.- It seems that the authors considered a 10-fold cross validation scheme to get unbiased accuracy estimates from the autoencoder models but don’t mention what they did for the 3D-CNN models. Since some kind of scheme should be used to obtain unbiased accuracies, this should be better explained por the 3D-CNN results.
6.- Related to sample sizes. Although very interesting, the fitting of multi input models with the three modalities is not too ambitious for such small sample size?
7.- Finally, in section 3.5 I do not understand which is the data used to carry out all these statistical comparisons between models. Do they come from 10-fold CV? This should be explained.

Author Response
The manuscript by Vyškovský applies two different deep learning approaches (stacked autoencoders and 3D- convolutional neural networks [3D-CNN]) to build classifiers for discriminating patients with a first episode of schizophrenia from healthy controls by means of three types of brain images derived from T1 MRI scans. The subject of the manuscript is of current interest for both the target population and the methods used. However, I have some methodological issues and doubts listed below.
Response: Dear reviewer, thank you for your valuable comments and insights, which enabled us to significantly improve our manuscript. We hope that we have responded to all your comments. Please see the attachment for revised version of the paper.
1.- Deep learning methods, due to the very large amount of parameters to be estimated, are typically applied to samples ranging from the hundreds to the millions (i.e. big data) where these methods excel. Here an N = 52 is used for each group. I acknowledge that recruiting first episode patients is difficult, but it I wonder if with such small samples the algorithms are just underperforming, and a “shallow” learning algorithm such as the SVM, random forest, GPC, lasso … would have equally o better performed. Including the results of one of them would enrich de manuscript.
Response 1: Thank you for your comment. In our previous paper, we also experimented with shallow classifiers such as SVM and not very deep multilayer perceptrons and their random subspace ensembles. The results were better but our goal was to compare feature extraction methods in combination with deep learning classifiers rather than to find the best performing classifiers. Thanks to your comment. We have added the following text to the discussion:
“The same dataset as in this paper was used in our previous work. The first paper [21] explored random subspace ensembles of multi-layer perceptrons (MLP) and SVMs, which were learned on the VBM and DBM data and their combination. The achieved accuracies were 73.12 % (MLP) and 73.51 % (SVM). And the second paper [20] presented the results on SVM trained on selected voxels using two-sample t-tests with an accuracy not exceeding 64 %. These results show that the results achieved in this paper are worse com-pared to shallow methods trained on the same dataset, and suggest that shallow methods can achieve better results with smaller sample sizes.”
2.- The author claim that both samples are matched for age, but what about gender? Since males and females differ in their T1 images, if the proportion of males is different in both samples the authors may have unknowingly fit classifiers for gender instead of disorder.
Response 2: We agree that the data should be matched for gender and our dataset consisted of only males as mentioned in paragraph 2.1 Schizophrenia Patients and Healthy Controls. We have corrected the text to emphasize the only males have been used for the analysis: „The entire dataset consisted of only male patients with median (min-max) age: SZ 22.9 (17–40), HC 23.0 (18.2–37.8).“
3.- Before fitting the autoencoders, the authors carried out a selection of most informative voxels through their significance in M-W tests. From the text, it may seem that they used the whole sample of patients and controls to make this previous selection of voxels. If this is the case, that would lead to clear positive biases in the estimates of validation accuracy (as data from validation individuals would have been used for the selection of voxels). This needs to be clarified.
Response 3: This information was indeed missing from the text. And we would like to ensure that the 10-fold cross-validation loop includes both feature selection and classification, so we have corrected the description of feature selection to make it clearer:
„After putting aside the testing samples, the Mann-Whitney test was applied to each voxel of the brain images – voxels at a specific location in each patient’s brain created one group, and voxels at the same location in healthy brains created a second group – and the application of the tests on the whole brain resulted in a probability map of p-values.“
4.- The authors carried out fittings for models with different number of layers and units (filters in CNN) as a kind of exploratory search for the best model, but how do they decide on the rest of hyperparameters? (number of epochs, size of batches, regularization, function types, …). Did they use the same data for selecting hyperparameters and evaluating the success of models?
Response 4: Neural networks are full of hyperparameters that need to be set. However, the computational time would be significantly longer if all parameters were set by a validation set, so we decided to explore only some of them as you mention. The other hyperparameters were set as follows. Regarding the 3D CNN, the number of epochs was controlled by the validation set to save computational time. On the other hand, the number of epochs in SAE was kept fixed. The batch size was 10, because with a larger number, some architectures tended to run out of memory, even a machine has 64 GB RAM. The rest of the hyperparameters, such as the regularization and activation functions, were left as default so as not to make the experiment too complex. The success of the models was evaluated on a test dataset that was the result of a 10-fold cross-validation. This dataset played no role in the selection of the hyperparameters.
5.- It seems that the authors considered a 10-fold cross validation scheme to get unbiased accuracy estimates from the autoencoder models but don’t mention what they did for the 3D-CNN models. Since some kind of scheme should be used to obtain unbiased accuracies, this should be better explained por the 3D-CNN results.
Response 5: The information was really missing from the paper, thank you for the notice. I have added it to paragraph 2.4 Experiment 2: 3D Convolutional Neural Network. Both the SAE and 3D CNN were validated in the same way:
„The training process was validated using 10-fold cross-validation in the same way as for SAEs…“
6.- Related to sample sizes. Although very interesting, the fitting of multi input models with the three modalities is not too ambitious for such small sample size?
Response 6: We combined the features to try and utilize all possibilities that the given dataset and feature extraction offered. However, the combination did not help, perhaps because it exacerbated the curse of dimensionality. Thank you for the comment; we have updated the text in the discussion:
„Input information combined from two or three data sources with the same sample size of patients and controls probably contributed to overfitting the models in the training phase only because it exacerbated the curse of dimensionality – too many features for too many subjects.“
7.- Finally, in section 3.5 I do not understand which is the data used to carry out all these statistical comparisons between models. Do they come from 10-fold CV? This should be explained.
Response 7: Thank you for this comment. The explanation was indeed missing from the paper and I have updated the text in paragraph 3.5 Statistical comparison to clarify what data entered the statistical analysis. The point is that all architectures were used repeatedly, which yielded multiple results that served as data for Mann-Whitney tests.
„Finally, the architectures with the best result for each dataset (T1, GMD, LVC) and models (SAE, 3D CNN) were statistically compared using the Mann-Whitney test. Accuracies that resulted from repeating all experiments 10 times were used as the data that entered the statistics, and the specific compared architectures were…“

Reviewer 2 Report

The main aim of the paper was to explore how different deep learning algorithms and image pre-processing steps could impact the classification of schizophrenia using MRI. The authors found that the approach based on the autoencoder performed better than CNN and the use of preprocessing steps as included in the VBM pipeline are those leading to the best accuracy.
The classification accuracies are in general quite low even if compared to the reference literature as mentioned by the authors in the introduction. Furthermore, it is not clear if and how their results on the preprocessing steps are generalizable.
Abstract:
Please add information on the dataset that was used to train the model and assess its performances. For example, some info about the type of images and their amount.
The aim of the paper must be described in the abstract.
Introduction:
What does BPD refer to?
Lines 64-68. Have the listed methods been applied in schizophrenia? If so, I would suggest providing more details on their performances. If not, I would suggest removing them from the introduction.
Introduction is too lengthy, and it contains a lot of information that can be removed. For example, the section about the basic principles of autoencoders and Convolutional Neural Networks must be considerably reduced or removed.
The research aim must be better described including the gaps that are currently present in the reference literature and how the paper intends to fill them.
Materials and Methods
The authors must better describe how the dataset was split into training/validation/test data. How many images were used to assess the accuracy of the approaches?
Since other methods using AI approaches were proposed to classify schizophrenia from morphological MR images [44-47], the authors must compare their results to those methods and justify why one of the focus of their work is put on the preprocessing. What do the other papers propose as pre-processing of MR images?
What does GMD refer to?
It is not clear how feature selection based on the Mann-Whitney test was performed.
The statistical approach must be better described including details o
Results
The quantitative results presented in Table 6 must be better described. It is not clear which parameters were compared. If all the parameters were included in the test a post hoc approach to highlight the differences between couples of values should be added.
The results reported in table 5 are not consistent with those described in table 4. Why didn’t the authors use the same CNN architecture?
In general, the results in terms of accuracy do not seem strong enough to justify the use of the methods proposed by the authors. In most of the conditions the accuracy is below 70%.
Discussion
The authors must discuss the reason why the VBM performs better than the others and analogously why the autoencoder is better than CNN.
Author Response
The main aim of the paper was to explore how different deep learning algorithms and image pre-processing steps could impact the classification of schizophrenia using MRI. The authors found that the approach based on the autoencoder performed better than CNN and the use of preprocessing steps as included in the VBM pipeline are those leading to the best accuracy.
The classification accuracies are in general quite low even if compared to the reference literature as mentioned by the authors in the introduction. Furthermore, it is not clear if and how their results on the preprocessing steps are generalizable.
Response: Dear reviewer, thank you for your valuable comments and insights that helped us to significantly improve our manuscript. We hope that we have responded to all your comments. Please see the attachment for revised version of the paper.
Abstract:
Please add information on the dataset that was used to train the model and assess its performances. For example, some info about the type of images and their amount.
The aim of the paper must be described in the abstract.
Response: Thank you for your suggestions to improve the abstract. We have updated the text to explicitly describe the main objective, and we have included information on dataset size and modality.
“…to teach them to classify 52 patients with schizophrenia and 52 healthy controls. The main aim of this study is to explore whether complex feature extraction methods can help improve the accuracy of deep learning-based classifiers compared to minimally preprocessed data.”
Introduction:
What does BPD refer to?
Response: Thank you for this notice. The abbreviation stands for Borderline Personality Disorder and we have written the full version because BPD is not used later in the paper.
Lines 64-68. Have the listed methods been applied in schizophrenia? If so, I would suggest providing more details on their performances. If not, I would suggest removing them from the introduction.
Response: The methods were used to detect differences between groups rather than to preprocess for classification, so we decided not to mention them in favor of shortening the introduction as you suggest in the following comment.
Introduction is too lengthy, and it contains a lot of information that can be removed. For example, the section about the basic principles of autoencoders and Convolutional Neural Networks must be considerably reduced or removed.
Response: We agree that the introductory section should be shorter and also the paragraphs with the basic principles of the two classifiers have been reduced to provide only the main idea of the SAE and 3D CNN and their advantages.
The research aim must be better described including the gaps that are currently present in the reference literature and how the paper intends to fill them.
Response: Thank you for this valuable comment, which significantly improves the clarification of our objective. We have added a penultimate paragraph in the introduction that explains what our goal is and how it relates to the methodology of published papers:
“Deep learning methods are state-of-art classifiers known for their ability to extract appropriate information for classification. However, morphometries as feature extraction methods used to play crucial role in brain preprocessing, where shallow classification methods were applied and enabled to prepare information for classifiers, and they improved the classification pipeline. Even nowadays, authors tend to use preprocessing [36]–[38], [40]. To the best of our knowledge, various morphometric methods have not yet been investigated and compared in combination with deep learning classifiers such as autoencoders and 3D CNN-based models, and the ability of these methods to properly ex-tract features alone has not been compared to morphometry preprocessing methods.”
Materials and Methods
The authors must better describe how the dataset was split into training/validation/test data. How many images were used to assess the accuracy of the approaches?
Response: Thank you for your advice on how to improve the explanation of the validation method. Based on the comment, we have updated the text to make it more understandable. Since we used a 10-fold cross-validation, each image occurred once in the test data. However, the validation set was only used to stop the 3D CNN training process. The updated parts of paragraphs are as follows:
„…the training process was validated using 10-fold cross-validation, where 10 % of the dataset was put aside in each fold before the features were selected and the network trained. After the last fold was performed, the complete testing data set consisted of 104 subjects, and each subject was classified by a network that was not trained on that subject to achieve unbiased results.“
„The training was controlled by the validation set, which consisted of 10 % of the training data and was stopped after 1000 epochs or when all of the following criteria were met: the minimal number of epochs was 100, accuracy on the validation set was decreasing, training accuracy was greater than 90 %, and validation accuracy exceeded 75 %.“
Since other methods using AI approaches were proposed to classify schizophrenia from morphological MR images [44-47], the authors must compare their results to those methods and justify why one of the focus of their work is put on the preprocessing. What do the other papers propose as pre-processing of MR images?
Response: The references [44-47] section (now rows 170-184) has been improved; we have added a brief description of the preprocessing methods used by other authors. The preprocessing steps vary from paper to paper, but the key steps such as registration, segmentation and smoothing are very common even when using deep networks. However, we believe that the impact of preprocessing has not been investigated in the range we have done. Rather, their valuable research focuses on other issues related to deep networks, such as dilatation, etc., and the added value lies elsewhere.
What does GMD refer to?
Response: The abbreviation stands for Gray Matter Density and we would like to kindly draw your attention to the introduction where the abbreviation is explained in its first occurrence.
It is not clear how feature selection based on the Mann-Whitney test was performed.
Response: The description of feature selection was not really well described, so we have updated the text and explained it in more  detail in 2.3 Experiment 1: Autoencoders. We would also like to reassure you that the feature selection was performed after the test data had been put aside, so as not to get biased feature selection results, as we were referred to by another reviewer pointing out that it was not described.
„After putting aside the testing samples, the Mann-Whitney test was applied to each voxel of the brain images – voxels at a specific location in each patient’s brain created one group, and voxels at the same location in healthy brains created a second group – and the application of the tests on the whole brain resulted in a probability map of p-values.“
The statistical approach must be better described including details o
Results
The quantitative results presented in Table 6 must be better described. It is not clear which parameters were compared. If all the parameters were included in the test a post hoc approach to highlight the differences between couples of values should be added.
Response: Thank you for these 2 comments. The explanation was indeed missing from the paper and I have updated the text in paragraph 3.5 Statistical comparison to clarify what data entered the statistical analysis. The point is that all the architectures were used repeatedly, which yielded multiple results – accuracies – that served as data for Mann-Whitney tests.
„Finally, the architectures with the best result for each dataset (T1, GMD, LVC) and models (SAE, 3D CNN) were statistically compared using the Mann-Whitney test. Accuracies that resulted from repeating all experiments 10 times were used as the data that entered the statistics, and the specific compared architectures…“
The results reported in table 5 are not consistent with those described in table 4. Why didn’t the authors use the same CNN architecture?
Response: Thank you for this comment, as the explanation was missing and Table 5 looked unintuitive. The reduced number of architectures explored was caused by the computational complexity of the task, because we were combining two or three 3D images, the computational time was very long, and the architectures with more than 7 layers even ran out of memory. We clarified our decision to reduce the number of architectures investigated in the text:
“Since learning the 3D CNN classifier based on the combination of the two or three features was computationally very intensive, we decided to explore only CNNs of 5 and 7 convolutional layers, which gave good results when single modality was used. Adding more layers was beyond our computational possibilities.”
In general, the results in terms of accuracy do not seem strong enough to justify the use of the methods proposed by the authors. In most of the conditions, the accuracy is below 70%.
Response: We understand your remark about the accuracy, however, the primary goal of our paper was not so much to recommend a classification method with an excellent accuracy as to compare different types of methods with respect to the input dataset used. We would also like to emphasize that the results presented in the article are averaged (run several times) to be independent of random weight initialization, although some of their runs have exceeded 68% accuracy. A small novelty of the discussion is also the fact that the dataset we use does not provide excellent results even when using traditional classifiers, however the shallow classifiers performed better.
“The same dataset as in this paper was used in our previous work. The first paper [21] explored random subspace ensembles of multi-layer perceptrons (MLP) and SVMs, which were learned on the VBM and DBM data and their combination. The achieved accuracies were 73.12 % (MLP) and 73.51 % (SVM). And the second paper [20] presented the results on SVM trained on selected voxels using two-sample t-tests with an accuracy not exceeding 64 %. These results show that the results achieved in this paper are worse compared to shallow methods trained on the same dataset, and suggest that shallow methods can achieve better results with smaller sample sizes.”
Discussion
The authors must discuss the reason why the VBM performs better than the others and analogously why the autoencoder is better than CNN.
Response: Thank you for the comment; we have added some ideas for why these methods might outperform others:
“The improvement may suggest that schizophrenia is more manifested in the gray matter, which is extracted from the brain in contrast with T1 or DBM, which is a whole-brain method.”
“On the other hand, the results were not as good as in the case of SAEs, suggesting that SAE can work more effectively with a smaller sample size, or that the preprocessing method of feature selection helped reduce noisy voxels from the images, or a combination of both.”
Round 2
Reviewer 1 Report
All issues addressed. Thanks.
Reviewer 2 Report
The authors addressed my comments and concerns.